# Possible Metastatic Stage-Dependent ILC2 Activation Induces Differential Functions of MDSCs through IL-13/IL-13Rα1 Signaling during the Progression of Breast Cancer Lung Metastasis

**DOI:** 10.3390/cancers14133267

**Published:** 2022-07-04

**Authors:** Atsushi Ito, Yuichi Akama, Naoko Satoh-Takayama, Kanako Saito, Takuma Kato, Eiji Kawamoto, Arong Gaowa, Eun Jeong Park, Motoshi Takao, Motomu Shimaoka

**Affiliations:** 1Department of Molecular Pathobiology and Cell Adhesion Biology, Mie University Graduate School of Medicine, Tsu 514-8507, Mie, Japan; y-akama@med.mie-u.ac.jp (Y.A.); a-2kawamoto@med.mie-u.ac.jp (E.K.); arong-g@doc.medic.mie-u.ac.jp (A.G.); epark@doc.medic.mie-u.ac.jp (E.J.P.); 2Department of Thoracic and Cardiovascular Surgery, Mie University Graduate School of Medicine, Tsu 514-8507, Mie, Japan; takao@med.mie-u.ac.jp; 3Department of Emergency and Disaster Medicine, Mie University Graduate School of Medicine, Tsu 514-8507, Mie, Japan; 4Laboratory for Intestinal Ecosystem, Center for Integrative Medical Sciences, RIKEN, 1-7-22 Suehiro-cho, Tsurumi-ku, Yokohama 230-0045, Kanagawa, Japan; naoko.satoh@riken.jp; 5Department of Hematology and Oncology, Mie University Graduate School of Medicine, Tsu 514-8507, Mie, Japan; kana-s@clin.medic.mie-u.ac.jp; 6Department of Cellular and Molecular Immunology, Mie University Graduate School of Medicine, Tsu 514-8507, Mie, Japan; katotaku@doc.medic.mie-u.ac.jp

**Keywords:** group 2 innate lymphoid cells, breast cancer, lung metastasis, micro- and macrometastasis, tumor microenvironment, myeloid-derived suppressor cells

## Abstract

**Simple Summary:**

When breast cancer metastasizes to the lung, group 2 innate lymphoid cells (ILC2s) are thought to promote tumor growth via the activation of myeloid-derived suppressor cells (MDSCs). In this study, we aimed to characterize the dynamic interactions of ILC2s and MDSCs during the course of cancer progression from the micrometastatic to the macrometastatic stages. We found that ILC2s were activated in both the micro- and macrometastatic regions, suggesting sustained activation throughout the metastatic cascades. In addition, our findings indicate that ILC2s may induce the immunosuppressive functions of MDSCs during the later stages of metastasis. Concomitantly, ILC2 may instigate extracellular matrix remodeling by polymorphonuclear (PMN)-MDSC activation during the early stages of metastasis. These metastatic-stage-specific changes may contribute to metastatic tumor growth in the microenvironment of breast cancer lung metastasis.

**Abstract:**

Breast cancer is the most common cancer in women worldwide, and lung metastasis is one of the most frequent distant metastases. When breast cancer metastasizes to the lung, group 2 innate lymphoid cells (ILC2s) are thought to promote tumor growth via the activation of myeloid-derived suppressor cells (MDSCs), which are known to negatively regulate anticancer immune responses. However, it remains to be elucidated exactly how this ILC2–MDSC interaction is involved in tumor growth during metastases formation. Using a 4T1/LM4 breast cancer mouse model, we found that ILC2s were activated in both the micro- and macrometastatic regions, suggesting sustained activation throughout the metastatic cascades via IL-33/ST2 signaling. Consistent with IL-13 secretion from activated ILC2s, the frequencies of polymorphonuclear (PMN)- and monocytic (M)-MDSCs were also significantly elevated during the progression from micro- to macrometastatic cancer. However, the effects of ILC2-induced MDSC functionality on the microenvironment differed in a metastatic-stage-specific manner. Our findings indicate that ILC2s may induce the immunosuppressive functions of MDSCs during the later stages of metastasis. Concomitantly, ILC2 may instigate extracellular matrix remodeling by PMN-MDSC activation during the early stages of metastasis. These metastatic-stage-specific changes may contribute to metastatic tumor growth in the microenvironment of breast cancer lung metastasis.

## 1. Introduction

Cancer immunotherapies have provided hope to patients with triple-negative breast cancer (TNBC) lung metastasis, an aggressive breast cancer expressing neither the estrogen receptor (ER), the progesterone receptor (PR), nor the human epidermal growth factor receptor 2 (HER2) [1,2]. Despite the success of immunotherapy in some patients, others experience no clinical benefits, which is often attributed to de novo resistance mediated by the tissue-specific tumor microenvironment (TME) [3]. The TME is comprised of various lymphoid and myeloid immune cells, mesenchymal-origin cells, and the extracellular matrix (ECM), all of which participate in tumor progression throughout all stages of tumorigenesis [4]. Since tissue-specific TME regulates tumor growth, determines metastatic progression, and impacts the therapeutic response [3,4,5], considerable attention in recent years has been focused on TME regulation in breast cancer lung metastasis. 

Lung metastasis consists of a complex and multistep process known as the invasion–metastasis cascade [6,7]. It begins with an epithelial–mesenchymal transition (EMT), which induces the reversible morphological and biochemical alterations that permit a specific epithelial cell to attain a mesenchymal phenotype. These alterations lead to decreased E-cadherin-mediated lateral adhesion of cells, as well as the loss of polarity and tight junctions, thereby dissociating the primary tumor into individual cells and resulting in an important growth advantage, which provides the basis for tumor spread [7,8]. In addition, EMT promotes the destruction of basement membranes and the endothelial barrier through the activation of matrix metalloproteinases (MMPs), thus facilitating tumor cell intravasation into the blood stream [9,10]. To succeed in metastatic colonization, circulating tumor cells must extravasate into the lung parenchyma and survive in the microenvironments of distant organs [11]. The microenvironment of a distant site usually differs greatly from the primary site of the tumor, both in terms of the types of stromal cells and in the composition of ECM. In addition, tumor cells are subject to immune surveillance by natural killer (NK) cells and CD8+ T-cells, which can target them for elimination. To overcome these unfavorable environmental factors, the primary tumor may secrete certain cytokines, chemokines, and exosomes into the circulation in order to reach the distant organ, wherein it forms pre-metastatic niches [2]. Niche formation involves the activation of immunosuppressive cells such as regulatory T cells and myeloid-derived suppressor cells (MDSCs), thereby providing a more hospitable environment for tumor cell survival [7]. In such microenvironments, extravasated tumor cells form small tumor nests (micrometastasis), some of which can further develop into clinically detectable lesions (macrometastasis) [12,13]. 

During this cancer progression from the micrometastatic stage to the macrometastatic stage, stromal cells such as fibroblasts, endothelial cells, and immune cells have been shown to drastically change their properties in the metastatic microenvironment [11,14,15,16]. Of these, the immunological alteration of the TME may play a central role in metastatic outgrowth. Metastatic tumor growth occurs when tumor cells evade immunological surveillance. Immunosuppressive cells, such as MDSCs, are known to negatively regulate anticancer immune surveillance in various types of cancers [17,18]. Thus, understanding the immune suppressive activity of MDSCs throughout the course of cancer progression is of crucial importance. However, very little is known about how MDSCs are activated and involved in tumor growth during the process of metastases formation.

MDSC activation in breast cancer has been shown to be induced by the IL-13 secreted from group 2 innate lymphoid cells (ILC2s) [19]. ILC2s, which lack adaptive antigen receptors, are the most recently described family of innate immune cells [20]. ILC2s express GATA3 and are characterized by the production of type 2 cytokines such as IL-4, IL-5, IL-13, and amphiregulin [21]. Whether or not ILC2s mediate pro- and/or antitumor effects has been shown to vary, depending on the tumor types and tissue-specific microenvironment [19,22,23,24,25]. In the context of breast cancer lung metastasis, ILC2s are believed to be deeply involved in forming the immune-suppressed microenvironment, generating both pre- and mature-metastatic niches. Studies using a 4T1 breast cancer mouse model have shown that ILC2-derived IL-13 promotes 4T1 lung metastasis via the activation of the IL-13Rα1 expressed in MDSCs [19,26]. Thus, previous studies suggest that the ILC2–MDSC axis may regulate tumor progression during breast cancer lung metastasis.

In this study, we set out to characterize the interactions of ILC2s with MDSCs during the course of cancer progression from the micrometastatic to the macrometastatic stages. To this end, we developed a 4T1/LM4 mammary carcinoma mouse model that showed the process of spontaneous metastasis to the lungs, thereby mimicking TNBC in humans. By analyzing the micrometastatic and macrometastatic lung regions in the 4T1/LM4 mammary carcinoma mouse model, we obtained snapshots of the temporal changes caused by ILC2 activation during the micrometastatic to macrometastatic stages. Here, we demonstrate how the potential interaction of ILC2s with MDSCs induce metastatic-stage-specific functional changes, contributing to metastatic tumor growth in breast cancer lung metastasis.

## 2. Materials and Methods

### 2.1. Mice

Balb/c female mice were purchased from Japan SLC, Inc. (Shizuoka, Japan) and maintained with a 12 h light/dark cycle at the Experimental Animal Facility of Mie University. Mice were age-matched whenever possible, and most were used at 8–12 weeks of age. The animal experimental procedures used in this study were approved by the Ethics Review Committee for Animal Experimentation at Mie University (No. 2020-12).

### 2.2. Development of the Mouse 4T1/LM4 Breast Cancer Model

The 4T1/luciferase murine mammary carcinoma cells used in this study were provided by Dr. Yoneda (Osaka University, Osaka, Japan) [27]. 4T1/luciferase cells were cultured in RPMI-1640 media (Nacalai, Kyoto, Japan) supplemented with 10% fetal bovine serum (FBS) (Equitech-Bio, Kerrville, TX, USA), 5 × 10^−5^ M2-mercaptoethanol, 100 U/mL penicillin, and 100 μg/mL streptomycin (Nacalai) in an incubator maintained at 37 °C with 5% CO_2_. By repeating in vivo passages of 4T1 cells, we established a subline designated 4T1/LM4, which was prone to metastasize to the lung. The 4T1/LM4 cells were cultured in sub-confluent monolayers during the log growth phase and were then harvested by brief treatment with 0.05% Trypsin-EDTA (Gibco), having been washed twice in PBS prior to injection. Then, 3 × 10^5^ 4T1/LM4 cells were suspended in 100 μL of PBS and injected subcutaneously into the fourth abdominal mammary gland of mice. Tumor growth was monitored and measured using Vernier calipers every 2 to 3 days. The tumor volume was then calculated according to the formula: V = (a^2^ × b)/2, where ‘a’ is the shortest transverse diameter and ‘b’ is the longest transverse diameter [28]. As with the previously described lung metastasis model [27,29,30], mice were sacrificed 21 days after tumor injection, and the primary tumors and lungs containing metastatic tumors were harvested. Metastatic tumor nodules on the lung surface were counted under visual inspection, after which the lungs were harvested for further processing. Lungs of tumor-free Balb/c female mice were used as controls. To investigate the temporal changes of immune cells in the TME, the lung tissues derived from this cancer model were divided in two regions: micrometastasis and macrometastasis. The macrometastatic regions contained macroscopically detectable tumors (i.e., larger than 2 mm), whereas the micrometastatic regions contained no macroscopically detectible tumors. Metastatic tumor foci were resected with scissors to spare as much of the surrounding lung tissues as possible. The surrounding lung tissues were restricted to less than 2 mm in diameter. In addition, micrometastatic regions contained the lung tissues of normal regions after any macroscopically detectible tumors in the same lobes had been eliminated. Thus, these were the remaining lung tissue samples following tumor resection.

### 2.3. Isolation of Leukocytes from the Lung

Lung tissues were minced into small pieces and twice incubated with 1 mg/mL collagenase (Fujifilm Wako Pure Chemical Corporation, Osaka, Japan) diluted in RPMI-1640 media for 45 min at 37 °C. Tissue pieces were removed with a 100 μm nylon mesh (Falcon, Durham, NC, USA) and the supernatant, including the leukocytes, was centrifuged. Next, 70% Percoll was layered onto the cells and then suspended with 40% Percoll (GE Healthcare, Stockholm, Sweden) and centrifuged at 2000 rpm at 25 °C for 20 min. The middle-layer after the Percoll gradient was treated with ammonium–chloride–potassium (ACK) lysing buffer (150 mM NH_4_Cl, 10 mM KHCO_3_, 0.1 mM EDTA) to remove the red blood cells. 

In the case of MDSC isolation, lung tissues were minced into small pieces and incubated with 1 mg/mL collagenase containing RPMI-1640 media for 60 min at 37 °C. Tissue pieces were removed with a 100 μm nylon mesh, and the supernatant was centrifuged. Red blood cells were removed using ACK lysing buffer. These single-cell suspensions were used for flow cytometric analysis of MDSCs. According to the manufacturer’s instructions for the MDSC isolation kit (Miltenyi Boitec, Bergisch-Gladbach, Germany), Ly6G^+^ cells were isolated as polymorphonuclear (PMN)-MDSCs, and Ly6G^-^Gr1^+^ cells were isolated as monocytic (M)-MDSCs using magnetic microbeads for RT-qPCR. 

The number of leucocytes contained in the micrometastatic and macrometastatic region samples was counted and normalized by using the sample weights.

### 2.4. Flow Cytometric Aanalysis

Cell suspensions were incubated with a combination of fluorescently conjugated monoclonal antibodies as follows: CD45.2- PerCP (104, BioLegend, San Diego, CA, USA), CD127- PE/Cy7 (DIH4, BioLegend), ICOS- APC (C398.4A, BioLegend), CD90.2- APC/Fire 750 (30-H12, BioLegend), PD-1- APC (29F.1A12, BioLegend), KLRG1- PE (2F1, BD Biosciences, San Jose, CA, USA), ST2- BV421 (U29-93, BD Biosciences), KLRG1- PE/Cy7 (2F1, BioLegend), Ly6C- FITC (HK1.4, BioLegend), Ly6G- APC/Fire 750 (1A8, BioLegend), CD11b- APC (M1/70, BioLegend), PD-L1- PE (MIH5, BD Biosciences) IL-5- PE (TRFK5, Thermo Fisher Scientific), IL-13- PE (eBio13A, Thermo Fisher Scientific). The lineage cocktail for the FITC-conjugated antibodies was as follows: CD3ε (145-2C11, BioLegend), CD4 (RM4-4, BioLegend), CD19 (1D3/CD19, BioLegend), CD11b (M1/70, BioLegend), CD11c (N418, BioLegend), TCRβ (H57-597, BioLegend), TCR γ/δ (GL3, BioLegend), Gr1 (RB6-8C5, BD Biosciences), NK1.1 (PK136, BioLegend), FcεRI (MAR-1, BioLegend), and Ter119 (Ter-119, BD Biosciences). ILC2s were identified using flow cytometry as Lin^-^ CD45.2^+^CD90.2^+^ST2^+^. MDSCs were subdivided into M-MDSCs and PMN-MDSCs. M-MDSCs were identified as CD45.2^+^CD11b^+^Ly6G^-^Ly6C^high^ and PMN-MDSCs were identified as CD45.2^+^CD11b^+^Ly6G^+^Ly6C^low^.

To analyze intracellular cytokine protein expression, cells were incubated with phorbol 12-myristate 13-acetate (50 ng/mL) and ionomycin (500 ng/mL) in the presence of GoldiStop (BD Biosciences) for 4 h. Intracellular staining was performed using an Intracellular Fixation and Permeabilization Buffer Set (Thermo Fisher Scientific, Waltham, MA, USA). Cells were analyzed using a Canto II flow cytometer (BD Biosciences). Positive signals were identified by comparing the staining with a fluorescence minus one control. Fc block (BD Biosciences) was used to prevent nonspecific antibody binding. Cell viability was determined using a Zombie Aqua Fixable Viability Kit (BioLegend, San Diego, CA, USA). Data were analyzed using FlowJo software.

### 2.5. RNA Isolation and Real-Time Quantitative PCR

Total RNA from lung tissues was isolated using TRIzol reagent (Thermo Fisher Scientific) according to the manufacturer’s instructions. Reverse transcription (RT) was performed using a Prime Script RT reagent kit (Takara Bio, Shiga, Japan). RNA purity (A260/A280 ratio) and concentrations were measured using Nanodrop2000. Quantitative real-time PCR was performed using SYBR Green (Thermo Fisher Scientific) with a StepOne system (Thermo Fisher Scientific) for measuring. β-actin mRNA served as an endogenous control to normalize mRNA levels using the 2^−ΔΔCT^ method. All primer sequences are listed in Table 1.

### 2.6. Immunofluorescence Staining

Lung tissues and primary tumors were fixed in 4% paraformaldehyde for 4 h and then embedded in OCT freezing media (Sakura Finetek, Tokyo, Japan). Subsequently, 8 μm sections were cut on a CM1850 cryostat (Leica Microsystems, Nussloch, Germany) and affixed to microscope slide glasses (Matsunami, Japan). Frozen sections were blocked with 0.1% Triton-X100 (Sigma), 1% BSA and 10% FBS containing 1× PBS solution at room temperature for 1 h. After washing with 1× PBS three times, the sections were stained with rat anti-mouse IL-33 monoclonal antibody (396118, Thermo Fisher) at 4 °C overnight. To detect the signal, goat Cy3-conjugated anti-rat immunoglobulin antibody (405408, BioLegend) was used. DAPI-stained slides were examined with a BZ-X 710 microscope (Keyence, Osaka, Japan) and analyzed using ImageJ software (imagej.nih.gov/ij/ accessed on 22 April 2022).

### 2.7. ELISA

The mouse blood was drawn by cardiac puncture and collected in heparinized tubes. The blood samples were then stored on ice before centrifuging at 2000× *g* for 10 min at 4 °C. Serum samples were collected in 0.5 mL tubes and stored at −80 °C until analysis. The concentrations of IL-33 in the mice serum samples were measured by enzyme-linked immunosorbent assay kits (Abcam, Japan) according to the manufacturer’s instruction. The absorbances at 450 nm were determined using a microplate reader (iMark, BIO-RAD Laboratories).

### 2.8. Statistical Analysis

Statistical analysis was performed using GraphPad Prism 8.0 (GraphPad Software, San Diego, CA, USA), and differences were considered statistically significant at *p* < 0.05. The values of the mean and standard error of the mean are provided. Survival rates were analyzed using the log-rank (Mantel–Cox) test. Statistical significance was obtained by using the unpaired t-test and ordinary one-way ANOVA; this was followed by a Tukey’s multiple comparisons test for multiple groups. Data represent the mean ± SEM.

## 3. Results

### 3.1. Separation of Metastases-Bearing Lung Tumor Samples into Micrometastatic and Macrometastatic Regions Using the 4T1/LM4 TNBC Mouse Model

We established an aggressive 4T1 subline designated 4T1/LM4, which was prone to metastasize to the lung. Balb/c female mice were injected orthotopically with 4T1/LM4 cells into the fourth abdominal mammary gland to develop the TNBC mouse model. The appearance of the tumor was monitored, and its diameter was measured every 2 to 3 days until the animal was sacrificed on day 21 after tumor injection. Approximately 20% of the tumor-bearing mice died by day 21 due to cancer progression (Figure 1A). The volume of primary tumors increased in a time-dependent manner (Figure 1B and Appendix A). Unlike the tumor-free mice, 4T1/LM4 tumor-bearing mice developed metastatic lung nodules (Figure 1C and Appendix A). To investigate the temporal changes of the immune cells in TME, we compared the micrometastatic and macrometastatic regions. We harvested lung tissues of both regions from 4T1/LM4 tumor-bearing mice and from the normal lungs of age-matched tumor-free Balb/c female mice, which were used as a control. As 4T1/LM4 cells, but not normal cells, express luciferase, the amount of breast cancer cells in each region was quantified by targeting the luciferase gene expression in 4T1/LM4, as previously reported [31]. As expected, luciferase was more highly expressed in the macrometastatic regions than in the micrometastatic regions. No expression of luciferase in the normal lung was detected (Figure 1D). Of note, these results substantiated the presence of tumor cells in the micrometastatic regions, where no lesions were macroscopically detectible. 

### 3.2. Lung ILC2s Remained Activated throughout the Metastatic Cascades, Thereby Producing IL-13

To investigate the effects of ILC2 on the metastatic microenvironment, we compared the ILC2 number and its frequency in the lymphocytes per whole lung tissue sample between tumor-free and 4T1/LM4 tumor-bearing mice. We regarded Lin^-^ CD45.2^+^ CD90.2^+^ ST2^+^ cells as mouse lung ILC2s [32]. Representative flow cytometry images of the lung ILC2 gating strategies between tumor-free and 4T1/LM4 tumor-bearing mice are shown in Appendix A. The number of ILC2s per whole lung tissue sample was significantly greater in 4T1/LM4 tumor-bearing mice compared to that in tumor-free mice (Figure 2A). However, there was no difference between the two groups in terms of the percentages of ILC2s in lymphocytes (Figure 2B). Subsequently, we analyzed the expression of IL-5 and IL-13 by ILC2s in the lung. Representative flow cytometry histograms of cytokine-producing lung ILC2s versus tumor-free and 4T1/LM4 tumor-bearing mice are shown in Appendix A. We found potent IL-13 production by ILC2s localizing in the lungs of 4T1/LM4 tumor-bearing mice. By contrast, IL-5 production was not altered by this microenvironmental change in the lungs of 4T1/LM4 tumor-bearing mice (Figure 2C). These results indicate that secretion of IL-13 was induced to a greater degree than that of IL-5 in the metastases-bearing lungs, suggesting that ILC2-derived IL-13 is involved in the immune response to the microenvironment of breast cancer lung metastasis.

To better characterize the trajectory of changes during metastases formation, we next assessed whether there was a difference in the ILC2 number and its frequencies between the micro-and macrometastatic regions. As ILC2s infiltrate the primary tumor, it was assumed that they might accumulate in the macrometastasis regions as has been previously reported [25,33]. Unexpectedly, we found no difference in the number of total lymphocytes and ILC2s per tissue weight among the four groups: normal lungs, 4T1/LM4 (whole lung), and micro- and macrometastatic regions (Figure 3A). In addition, the percentages of ILC2s in lymphocytes also showed no difference among the four groups (Figure 3B). However, as tumor-infiltrating immune cells greatly altered their activity and phenotype—e.g., tumor -associated macrophages and regulatory T cells—we investigated whether there were any differences in the activation status of ILC2s in the micro- versus the macrometastatic regions. 

To determine whether ILC2s were preferentially activated by microenvironmental changes, we examined the suppressed expression levels of tumorgenicity 2 (ST2) and inducible T-cell costimulator (ICOS), which are known to be involved in ILC2 activation in the lung. It is well known that ST2, a receptor of IL-33, participates in the recruitment and activity of ILC2s throughout the IL-33/ST2 signaling pathway, thereby promoting IL-5 and IL-13 production [34]. ICOS plays a pivotal role in T-cell survival and function [35], and the binding of ICOS to ICOS ligand activates a cascade of intracellular signaling molecules that prevents apoptosis and leads to the production of cytokines [36]. In our tumor-bearing mice, the expression of ST2 and ICOS on ILC2s was significantly elevated in both the micro- and macrometastatic regions compared to normal lungs. The expression of ST2 and ICOS was most elevated in the macrometastatic regions, suggesting that ILC2s were most activated during macrometastasis (Figure 3C). A similar trend was observed in the expression of killer cell lectin-like receptor G1 (KLRG1), a well-conserved transmembrane C-type lectin receptor [32,37], although its heightened levels were still lower than those of the other receptors, ST2 and ICOS (Figure 3C). 

Programmed cell death 1 (PD-1) plays a crucial role as an immune checkpoint and displays tumor regression, indicating a rebalancing of the immune responses. We, therefore, evaluated the expression of PD-1 in order to gauge the impact on ILC2s’ regulation. PD-1 expression on ILC2s tended to increase during all stages of lung metastasis, albeit not to a statistically significant degree (Figure 3D). In support of this observation, immunofluorescence demonstrated that CD3^-^CD127^+^PD-1^+^ cells, which indicated the presence of PD-1^+^ ILCs, were more abundant in the lungs of tumor-bearing mice than in tumor-free mice (Appendix A). Taken together, these results suggest that ILC2s are activated in a stage-dependent manner throughout the metastatic cascades, although it remains uncertain whether PD-1 expression on ILC2 is an inhibitory response that occurs relative to cancer-associated ILC2 activation.

### 3.3. Stage-Specific Gene Signatures in the Pulmonary TME throughout the Metastatic Cascades

Aiming to delineate the stage-specific gene signatures throughout the metastatic cascades, we further investigated which lung tissues were affected by the induction of IL-13-producing ILC2. Consistent with the flow cytometric results showing the activation of ILC2, we found that IL-13 expression (though IL-5) was significantly elevated in macrometastatic regions compared to normal lung and micrometastatic regions (Figure 4).

MMP-9 is a zinc-dependent peptidase that belongs to the gelatinase subfamily of MMPs [38], causing degradation of gelatin and collagens during ECM remodeling in both tumor invasion and metastasis [39]. Therefore, we next assessed MMP-9 expression in both the micro- and macrometastatic regions of the lung and compared them with that of the normal lung. Interestingly, we observed that MMP-9 expression was strongly induced in the micrometastatic regions, whereas it was lower in the macrometastatic regions (Figure 4). Since it has been reported that IL-1β acts as a trigger when MMP-9 induction occurs through the activation of the mitogen-activated protein kinase (MAPK) pathway and the translocation of nuclear factor-κB (NF-κB) in cancer cells [40,41] and fibroblasts [42], we next assessed the expression of IL-1β in each region of the lung. As expected, IL-1β expression was highly upregulated in micrometastatic regions but was lower in macrometastatic regions, similar to MMP-9 (Figure 4). These findings indicate that IL-1β may stimulate tumor cells to produce MMP-9 during micrometastasis, thereby leading to ECM remodeling and the establishment of a favorable microenvironment for tumor growth. Furthermore, the IL-1β-induced ECM remodeling programs might be induced more robustly during the micrometastatic stage than during the macrometastatic stage, suggesting that the pulmonary TME is altered in a stage-specific manner.

### 3.4. Fluorescence Intensity of IL-33 Was Elevated Not Only in the Metastatses-Bearing Lungs but Also in the Primary Mammary Tumors

As ILC2s are activated thorough IL-33/ST2 signaling [32,43,44], we next sought to assess the source of endogenous IL-33 in 4T1/LM4 tumor-bearing mice. Immunofluorescence analysis demonstrated that IL-33 expression was observed in both the metastases-bearing lungs and the primary mammary tumors but not in the normal lungs (Figure 5A). In addition, the fluorescence intensity of IL-33 was elevated in the metastases-bearing lungs and even more so in the primary mammary tumors (Figure 5B). These results suggest that IL-33 was secreted not only from the metastatic sites but also from the primary tumor sites.

### 3.5. ILC2 Activation Is Associated with the Induction of Differential MDSC Functions during Cancer Progression from the Micro- to the Macrometastatic Stage

MDSCs express IL-13Rα1 and are activated by the IL-13 produced from ILC2s [19,23,45]. As IL-13 production from ILC2s in the lung was highly upregulated in the macrometastatic regions, we next assessed whether MDSCs were directly associated with IL-13-producing ILC2s for immune regulation in the lung metastasis. 

MDSCs contain two major subsets: PMN-MDSCs and M-MDSCs. The former share many of the morphological and phenotypic characteristics of neutrophils, whereas M-MDSCs are similar to monocytes [18]. PMN-MDSCs and M-MDSCs are defined as CD11b^+^Ly6G^+^Ly6C^low^ and CD11b^+^Ly6G^-^Ly6C^high^ cells, respectively. However, these markers are present on multiple immune cells, thus requiring careful exclusion gating for stringent flow cytometric identification of the MDSC subsets. Accordingly, in our study, a certain population of MDSCs was determined by using CD45.2, a hematopoietic marker, following the previously reported panels [46] (Appendix A). 

In accordance with this definition of MDSCs, PMN-MDSCs and M-MDSCs were purified as Ly6G^+^ and Ly6G^-^Gr1^+^ cells for analysis from normal lungs and from lung tissues containing the micro- or macrometastatic regions. The percentages of PMN-MDSCs and M-MDSCs in CD45.2+ cells were significantly increased in both the micro- and macrometastatic regions (Figure 6A). We further analyzed the expression of programmed cell death ligand 1 (PD-L1), demonstrating that the expression of PD-L1 on M-MDSCs was more clearly elevated in the macrometastatic regions than that of PMN-MDSCs (Figure 6B). As PD-L1 is one of the important immune regulatory molecules of M-MDSCs [47], it may suppress antitumor immunity via PD-1/PD-L1 signaling in the macrometastatic regions.

MDSCs mediate immune suppression through arginine depletion by expressing the enzyme arginase (Arg). In addition, they mediate nitric oxide (NO) production by expressing inducible nitric oxide synthase (iNOS) [48,49]. To determine whether the functionality of lung MDSCs differs in the micro- versus macrometastatic regions, we next assessed the gene expression of Ly6G^+^ and Ly6G^-^Gr1^+^ cells isolated from normal lungs and from lung tissues containing micro- or macrometastatic regions. qRT-PCR results confirmed that Arg1, the arginase-related gene common to both Ly6G^+^ and Ly6G^-^Gr1^+^ cells, was upregulated in the macrometastatic regions, whereas iNOS expression did not significantly change in either the micro- or macrometastatic regions (Figure 6C). In conjunction with the elevated expression of IL-13 expression in the macrometastatic regions (Figure 4), the immunosuppressive activity of MDSCs may be induced by Arg1, rather than iNOS, through IL-13/IL-13Rα1 signaling. Not only was IL-13Rα1 expression of Ly6G^+^ cells upregulated in the micrometastatic regions but that of Ly6G^-^Gr1^+^ cells was also similarly upregulated in both the micro- and macrometastatic regions (Figure 6C). It is worth noting that the gene expression of IL-1β was upregulated in Ly6G^+^ cells in the micrometastatic regions, as was IL-13Rα1 expression (Figure 6C), which may be responsible for the elevated expression of IL-1β observed in ECM remodeling of the micrometastatic regions (Figure 3E).

Taken together, our findings suggest that stage-dependent ILC2 activation might induce differential functioning of MDSCs through IL-13/IL-13Rα1 signaling during cancer progression from micro- to macrometastasis in breast cancer lung metastasis. ECM remodeling appears to be instigated by PMN-MDSC activation during the early stages of metastasis, while the immunosuppressive functions of PMN- and M-MDSCs are induced during the late stages of metastasis (Figure 7).

## 4. Discussion

In this study, we found that ILC2s were activated by ST2 and ICOS upregulation in both the micro- and macrometastatic regions, thereby supporting the possibility that ILC2s are activated throughout the metastatic cascades of breast cancer cells in the lung. Furthermore, we confirmed that IL-33 was released from both the metastases-bearing lungs and the primary mammary tumors, which might induce ILC2 activation. IL-13 expression was upregulated only in the macrometastatic regions, whereas IL-1β and MMP-9 expressions were upregulated only in the micrometastatic regions. As ILC2s activate MDSCs through IL-13/IL-13Rα1 signaling, we sought to determine whether MDSCs were directly associated with IL-13-producing ILC2s for immune regulation in lung metastasis. Interestingly, the frequencies of PMN- and M-MDSCs were also significantly increased in both the micro- and macrometastatic regions, thereby supporting the possibility that they are activated throughout the metastatic cascades of breast cancer cells in the lung. However, ILC2s induced their functions on the microenvironment in a stage-specific manner. We found that PD-L1 expression on M-MDSC and Arg1 expression of both PMN- and M-MDSCs were increased during macrometastasis. As PD-L1 and Arg1 are closely related to the inhibition of cytotoxic T-cell proliferation and activation, ILC2s may induce the immunosuppressive functions of MDSCs during the late stage of metastasis. In contrast, IL-1β expression of PMN-MDSCs was most elevated during micrometastasis. Since IL-1β is known to promote MMP-9 production in tumor cells, ILC2 may instigate ECM remodeling by PMN-MDSC activation during the early stages of metastasis.

IL-33 is mainly produced by epithelial cells and can activate ILC2s through the ST2 receptor, thereby prompting additional immune responses [32,43,44]. Unlike IL-1β and IL-18, the activation of IL-33 does not require cleavage by caspases; rather, IL-33 is likely to be released from necrotic cells as a full-length active molecule [50]. In our study, the fluorescence intensity of IL-33 was significantly elevated in the lungs and primary mammary tumors of 4T1/LM4 tumor-bearing mice but not in the lungs of tumor-free mice (Figure 5). These results indicate that IL-33 is released from necrotic cells within the tumor tissues. However, the gene expression of IL-33 showed no difference between the lungs of tumor-free mice and tumor-bearing mice, whereas there was a trend toward higher gene expression of IL-33 in the primary mammary tumors (Appendix A). Based on these results, we assessed IL-33 expression using the luciferase gene, which is 4T1/LM4 cell-specific, as a loading control and found that IL-33 expression was higher in metastases-bearing lungs compared to that in the primary mammary tumors (Appendix A). However, this result requires some interpretation: a previous study using a 4T1 mammary tumor mouse model showed that 4T1 cells are not the main source of IL-33 within tumor tissues [51]. IL-33 is produced from epithelial cells, endothelial cells, and tumor-associated fibroblasts within the TME [51]. Furthermore, luciferase expression was lower in the metastases-bearing lungs than in the primary mammary tumors. Accordingly, it may not be appropriate to use the 4T1/LM4-cell-specific luciferase gene as a loading control. Serum IL-33 concentration in tumor-bearing mice tended to be higher than that in tumor-free mice though not to a statistically significant degree (Appendix A). Therefore, we suggest the possibility that primary mammary tumor–derived IL-33 may reach the lung through the bloodstream, promoting the proliferation and activation of ILC2s. Consistent with our idea, previous reports have shown that intratumor gene expression and the serum protein of IL-33 increased with primary tumor growth in the 4T1 breast cancer mouse model [19,51].

Consistent with elevated levels of IL-33, the ILC2 number per whole lung tissue was found to be higher in the 4T1/LM4 tumor-bearing mice than in the tumor-free mice (Figure 2A). However, as the number of lymphocytes was also higher in the tumor-bearing mice, the percentages of ILC2s in lymphocytes showed no difference among the four groups (Figure 2B). We found no changes in the normalized number of ILC2s among the four groups: normal lungs, 4T1/LM4 (whole lungs), and micro- and macrometastatic regions (Figure 3A). These results could be attributed to the increased weight of tumor tissues in metastases-bearing lungs compared to normal lungs (Appendix A).

Although there was no change in the number or frequency of ILC2s during the different metastatic stages, the results we provide herein support differential ILC2 activation during the course of cancer progression from micro- to macrometastasis. We have shown that the expression levels of ST2, ICOS, and KLRG1 on ILC2s were elevated in the micrometastatic regions and were even more elevated in the macrometastatic regions (Figure 3C). 

In general, high KLRG1 expression correlates with low proliferative capacity and increased apoptosis in both NK cells and T-cells [37]. However, our recent study suggested the intriguing possibility that KLRG1 may be involved in the activation, but not the inactivation, of ILC2s in septic mice models [32]. Therefore, elevated expression of ST2, ICOS, and KLRG1 was thought to correlate with the activation of ILC2s during the process of lung metastasis. These results suggest that ILC2s are activated throughout the metastatic cascades while upregulating IL-13 in a stage-dependent manner. The expression of PD-1 on ILC2s was observed throughout all stages of lung metastasis (Figure 3D). Of note, PD-1 expression tended to increase in the macrometastatic regions. These findings are consistent with previous reports showing that tumor-infiltrating ILC2s highly express PD-1 in malignant melanomas [25], as well as in a pancreatic carcinoma model [33]. As IL-33 has been shown to induce ILC2 proliferation and enhance cytokine production, in conjunction with increased PD-1 expression [25,33], our results documenting increased PD-1 on ILC2s in the metastases-bearing lung tissues could indicate the suppression of ILC2 activation. In a malignant melanoma, ILC2-derived granulocyte macrophage–colony stimulating factor (GM-CSF) induced tumor-infiltrating eosinophils and exerted antitumor effects, which were exacerbated by antibody-mediated inhibition of PD-1 on ILC2s [25]. In pancreatic carcinomas, IL-33 drives ILC2 activation, which promotes CD103+ dendric cell recruitment through the production of CCL-5 and, consequently, CD8+ T cell activation [33]. Interestingly, the combination of IL-33 and the PD-1 antibody showed synergistic antitumor effects on melanoma and pancreatic cancer [25,33]. However, it remains unclear whether activating ILC2s would mediate similar antitumor effects in lung metastasis of breast cancer, since MDSCs are activated via IL-13/IL-13Rα1 signaling to induce pro-tumor effects [19,23,26].

The immunosuppressive activity of MDSCs, such as Arg1 and iNOS, has been shown to be enhanced by interferon (IFN)-γ, IL-13, and GM-CSF [52]. Consistent with elevated IL-13 expression in the macrometastatic regions (Figure 4), Arg1, but not iNOS expression, was elevated in both PMN- and M-MDSCs (Figure 6C). Here, we speculate that the immunosuppressive activity of MDSCs was induced by Arg1, rather than iNOS, through IL-13/IL-13Rα1 signaling in breast cancer lung metastasis. However, as ILC2s are known to also produce GM-CSF [25], ILC2-derived GM-CSFs may be involved in the activation of Arg1 in MDSCs. On the other hand, IL-1β in PMN-MDSCs was upregulated in the micrometastatic regions, in tandem with the elevated expression of IL-13Rα1 (Figure 6C). IL-1β stimulation of tumor cells activates multiple signaling pathways involving protein kinase B, MAPK, and NF-κB [41]. These signaling molecules have been reported to promote the production of MMP-9 from tumor cells, leading to the degradation of the ECM and the subsequent adhesion of disseminated tumor cells [40,41]. Therefore, we speculate that PMN-MDSC-derived IL-1β may stimulate tumor cells to produce MMP-9 in micrometastatic regions, thereby leading to ECM remodeling and the establishment of a favorable microenvironment for tumor growth. Taken together, our findings suggest the potential interaction of ILC2s with MDSCs leading to metastatic tumor growth during different stages of breast cancer lung metastasis.

One major technical limitation of the present study concerns the assessment of temporal changes in immune cells in TME, which forced us to divide the metastases-bearing lungs into two categories: tumor foci larger than 2 mm were classified as macrometastases and adjacent normal lung tissues as micrometastases. Then, we compared micro- and macrometastasis at the same time during the late stage (i.e., spatial-location-dependent separation). On the other hand, previous studies compared them by analyzing each mouse separately over the time course following tumor implantation (i.e., time-course-dependent separation) [11,53,54]. These previous studies then defined the early stage as micrometastasis and the late stage as macrometastasis, separately, over the time course following tumor implantation in order to assess the differences between ECM remodeling, angiogenesis, and fibroblast proliferation [11,53,54]. Using a different approach, we sought to study the potential interaction of ILC2s with MDSCs during the late stage, in which micro- and macrometastasis regions simultaneously exist in the lung. The comparisons between micro- and macrometastasis regions in the late stage may be of biological significance, possibly recapturing some changes in ILC2–MDSC interactions in the TME during the spatiotemporal progression of lung metastasis. However, our results may be susceptible to a tumorigenesis-related bias: regions that remained micrometastatic until the late stage may simply reflect slow tumor growth due to an unfavorable TME. In addition, it is not certain when the micrometastases were formed in the lung during these late stages. Despite the differences in interpretation of the study results between our method and those of previous studies, both methods could only capture a certain snapshot in the transition of breast cancer lung metastasis. However, the methods currently available for systematically analyzing dynamic metastatic cascades remain quite challenging. They require the use of new technologies such as in vivo live imaging to simultaneously identify each of the various cells within the TME.

Another possible drawback concerns our inability to unravel the causal relationship between the augmented interaction of ILC2 with MDSC and the growth of metastatic tumor foci from the micro- to the macrometastatic stages. ILC2 activation may simply be occurring as a result of the formation of lung metastasis. To clarify the above, we need a more detailed understanding of the trajectory of immunomodulation during the formation of lung metastasis, which may be made possible by recent advanced technologies. For example, three-dimensional (3D) culture systems of cancer cells combined with several biomaterials enable one to assess cancer invasion, migration, proliferation, and even the effects of anticancer drugs in more-realistic cancer environments under in vitro condition [55]. In addition, a multi-organ microfluidic chip that consists of two bionic organ units can recapitulate the blood–brain barrier, allowing even the evaluation of brain metastasis [56]. In vivo experiments using multiple transgenic mouse models have shown spontaneous development of breast cancer and lung metastases, as well as EMT expression markers of metastasized tumors by using a cell lineage tracing approach [57]. Furthermore, a series of recent advances in tissue-clearing-based 3D imaging may provide important breakthroughs for visualizing the overall dynamics of the metastatic progression and spatiotemporal distribution of individual cells in the metastatic lung TME [58]. These advanced technologies are expected to elucidate the mechanism underlying immunomodulation through the ILC2–MDSC interactions in the TME of breast cancer lung metastasis, thereby leading to the development of further immunotherapies.

## 5. Conclusions

Here, we demonstrated that metastatic-stage-specific immunomodulation may contribute to lung metastatic tumor growth in breast cancer lung metastasis. Our findings have the potential to provide new insights into the interaction of ILC2s with MDSCs in the lung TME, which could serve as the foundation for novel immunotherapies to treat breast cancer lung metastasis.

## Figures and Tables

**Figure 1 cancers-14-03267-f001:**
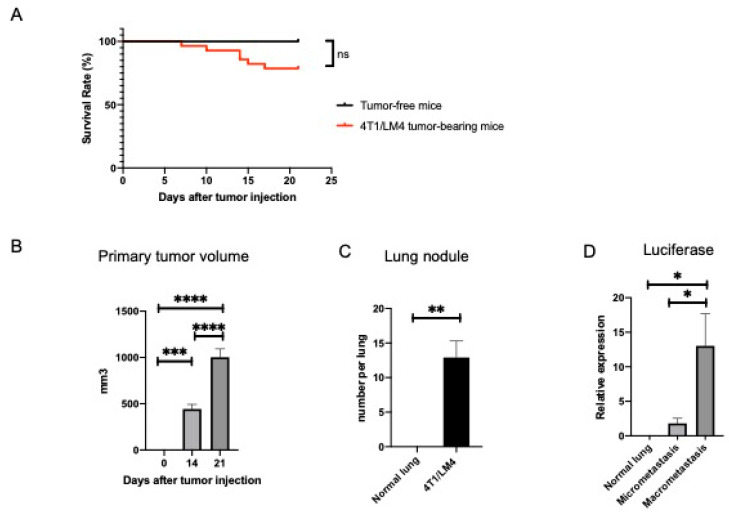
Characterization of a mammary carcinoma mouse model using 4T1/LM4 cells, which were prone to metastasize to the lung. (**A**) Survival rates were monitored and expressed as a log-rank test survival curve (tumor-free mice: *n* = 12, 4T1/LM4 tumor-bearing mice: *n* = 28). (**B**) Primary tumor volume constantly increased in a time-dependent manner (4T1/LM4 tumor-bearing mice: *n* = 10). (**C**) Lung nodules were observed at day 21 after 4T1/LM4 tumor injection. The number of metastatic nodules per entire lung was counted under visual inspection (normal lung: *n* = 12, 4T1/LM4 metastases-bearing lung: *n* = 10). (**D**) RT-qPCR analysis for gene expression of luciferase is shown. Total RNA was isolated from the normal lungs of tumor-free mice, as well as from the micro- and macrometastatic regions of 4T1/LM4 tumor-bearing mice (normal lung: *n* = 4, micrometastasis: *n* = 5, macrometastasis: *n* = 5). In (**A**), ns indicates not significant. * *p* < 0.05. ** *p* < 0.01. *** *p* < 0.001. **** *p* < 0.0001.

**Figure 2 cancers-14-03267-f002:**
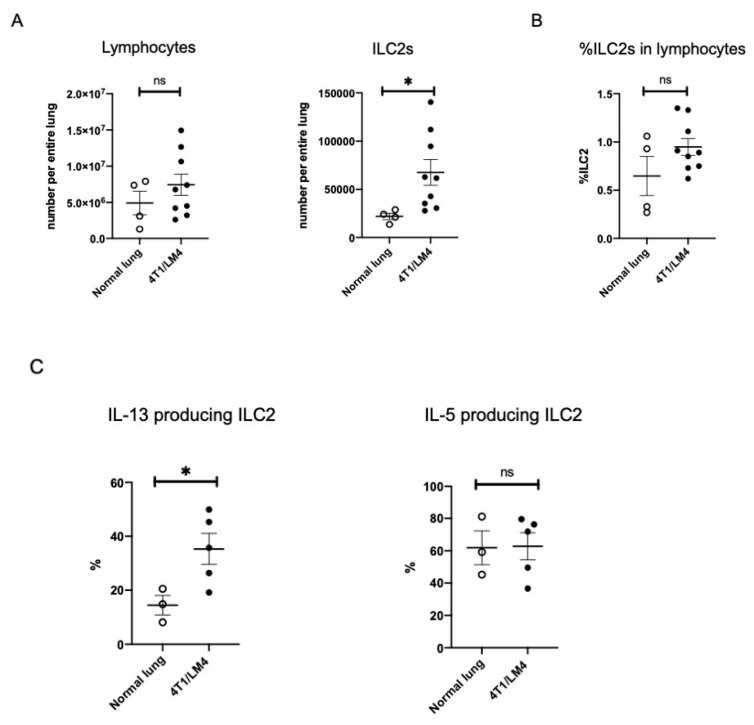
Comparison of the number, frequency, and cytokine production of ILC2s in the lungs of tumor-free and 4T1/LM4 tumor-bearing mice. (**A**) Absolute number of lymphocytes and ILC2s per entire lungs and (**B**) the percentages of ILC2s in lymphocytes are shown in the lungs of tumor-free and 4T1/LM4 tumor-bearing mice (normal lung: *n* = 4, 4T1/LM4: *n* = 9). (**C**) Percentages of IL-13- and IL-5-producing ILC2s in the lungs of tumor-free and 4T1/LM4 tumor-bearing mice are shown (normal lung: *n* = 3, 4T1/LM4: *n* = 5). The data were compiled from two or three independent experiments. The data from control and tumor-bearing mice were evaluated using the unpaired t-test. Here, ns indicates not significant. * *p* < 0.05.

**Figure 3 cancers-14-03267-f003:**
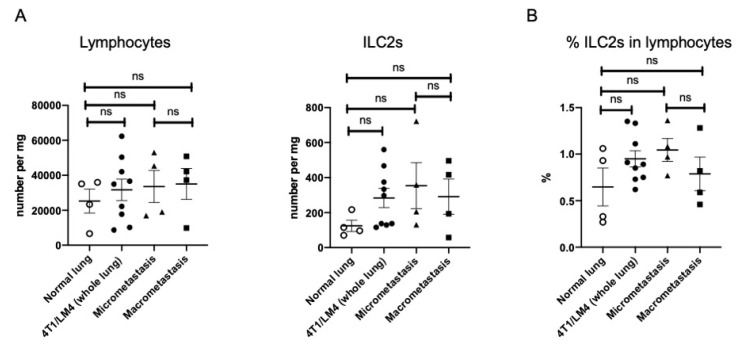
Comparisons of the number, frequency, and activity of ILC2s between normal lungs, whole lungs of tumor-bearing mice, and micro- and macrometastatic regions. (**A**) Normalized numbers of lymphocytes and ILC2s, (**B**) percentages of ILC2s in lymphocytes, (**C**) mean fluorescence intensities of ST2, ICOS, KLRG-1, and (**D**) PD-1 expressed on ILC2 are shown (normal: *n* = 4, 4T1/LM4 (whole lung): *n* = 9, micrometastasis: *n* = 4, macrometastasis: *n* = 4). The data were compiled from two or three independent experiments. ns, not significant; ILC2s, group 2 innate lymphoid cells; ST2, suppression of tumorgenicity 2; ICOS, inducible T-cell costimulatory; KLRG1, killer cell lectin-like receptor G1; PD-1, programmed cell death 1. * *p* < 0.05. ** *p* < 0.01. *** *p* < 0.001.

**Figure 4 cancers-14-03267-f004:**
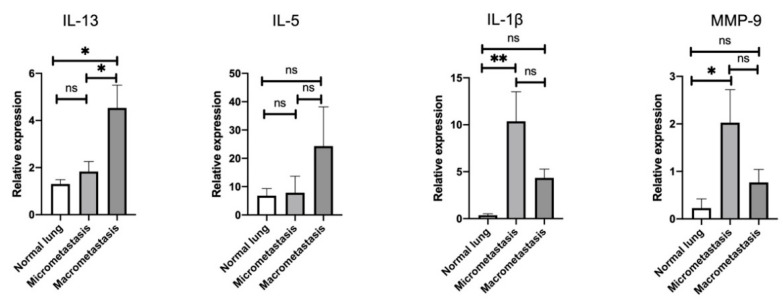
mRNA expression levels of IL-13, IL-5, IL-1β, and MMP-9. The micro- and macrometastatic regions of tumor-bearing mice were determined by RT-qPCR (normal lung: *n* = 4–6, micrometastasis: *n* = 4–6, macrometastasis: *n* = 4–6). The data were compiled from two or three independent experiments. ns, not significant; MMP-9, matrix metalloproteinase-9. * *p* < 0.05. ** *p* < 0.01.

**Figure 5 cancers-14-03267-f005:**
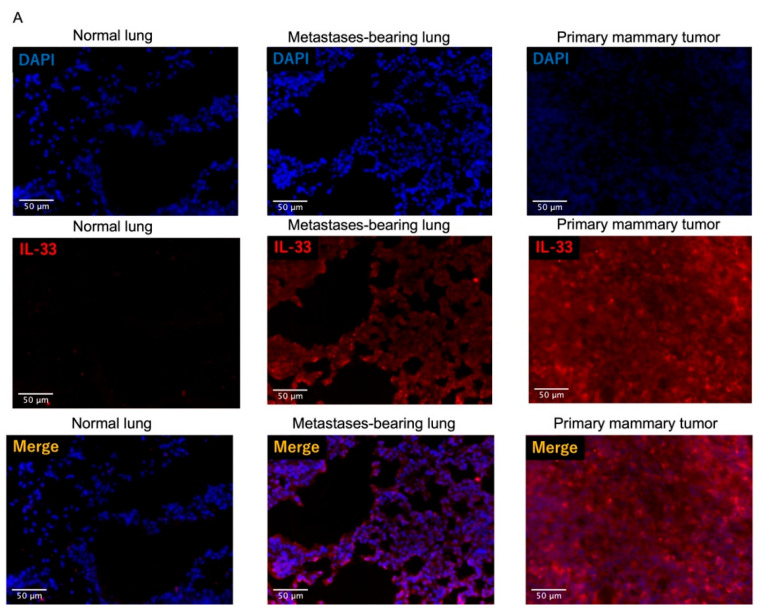
Analysis of fluorescence intensity of IL-33 between normal lungs, metastases-bearing lungs, and primary mammary tumors. (**A**) Frozen sections of the lungs of control mice, metastases-bearing lungs, and the primary mammary tumors were stained with anti-IL-33 antibodies (red) and DAPI (blue). (**B**) Relative expression levels of IL-33 between the normal lungs of tumor-free mice, the metastases-bearing lungs, and the primary mammary tumors were assessed as the mean light intensity normalized with DAPI. Six tissue sections per group (*n* = 2 mice per group) were imaged and quantified using Image J analysis software. ns, not significant. * *p* < 0.05. *** *p* < 0.001.

**Figure 6 cancers-14-03267-f006:**
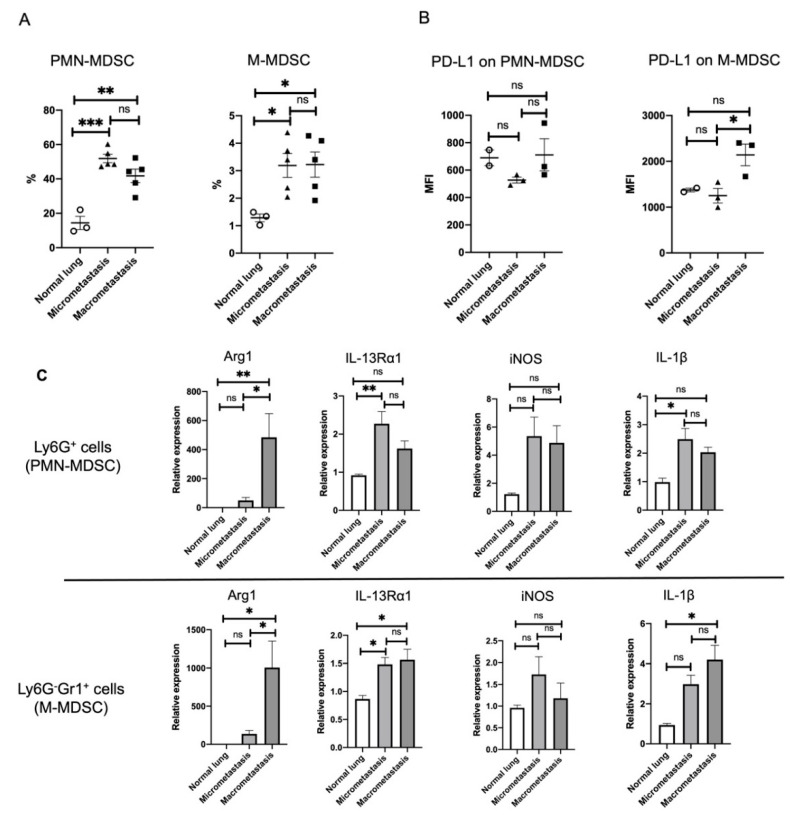
Comparison of frequencies and activities of MDSCs between normal lungs and micro- and macrometastatic regions. (**A**) Percentage of MDSCs in CD45.2+ cells, (**B**) mean fluorescence intensity (MFI) of PD-L1 expression on MDSCs between the normal lungs of tumor-free mice, and the micro- and macrometastatic regions of tumor-bearing mice (normal lung: *n* = 2–3, micrometastasis: *n* = 3–5, macrometastasis: *n* = 3–5). (**C**) PMN-MDSCs and M-MDSCs were isolated as Ly6G^+^ cells and Ly6G^-^Gr1^+^ cells, respectively, from the micro- and macrometastatic regions of tumor-bearing mice using magnetic microbeads. As a control, Ly6G^+^ cells and Ly6G^-^Gr1^+^ cells from the normal lungs of tumor-free mice were also isolated. The mRNA expression levels of Arg1, IL-13Rα1, iNOS, and IL-1β in isolated cells from each region were determined by RT-qPCR (normal lung: *n* = 4, micrometastasis: *n* = 5–6, macrometastasis: *n* = 5–6). ns, not significant; M-MDSC, monocytic myeloid-derived suppressor cell; PMN-MDSC, polymorphonuclear myeloid-derived suppressor cell; PD-L1, programmed cell death ligand 1; Arg1, arginase 1; IL-13Rα1, IL-13 receptor α 1; iNOS, inducible nitric oxide synthase. * *p* < 0.05. ** *p* < 0.01. *** *p* < 0.001.

**Figure 7 cancers-14-03267-f007:**
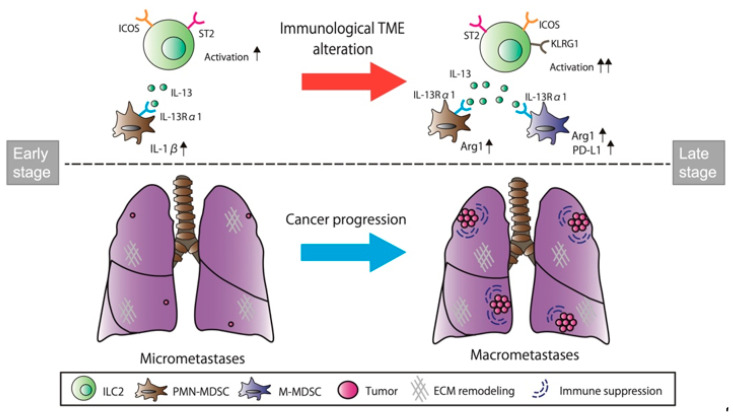
Stage-dependent ILC2 activation induces differential functions of MDSCs through IL-13/IL-13Rα1 signaling during cancer progression from micro- to macrometastasis. ILC2s, group 2 innate lymphoid cells; ST2, suppression of tumorgenicity 2; ICOS, inducible T-cell costimulatory; KLRG1, killer cell lectin-like receptor G1; M-MDSC, monocytic myeloid-derived suppressor cell; PMN-MDSC, polymorphonuclear myeloid-derived suppressor cell; PD-L1, programmed cell death ligand 1; Arg1, arginase 1; IL-13Rα1, IL-13 receptor α1.

**Table 1 cancers-14-03267-t001:** Primer sequences.

Primers	Forward 5′–3′	Reverse 5′–3′	Species
β-actin	CATCGTACTCCTGCTTGCTG	AGCGCAAGTACTCTGTGTGG	mouse
IL-13	GCTTATTGAGGAGCTGAGCAAC	GGCCAGGTCCACACTCCATA	mouse
IL-5	CGCTCACCGAGCTCTGTTG	CCAATGCATAGCTGGTGATTTT	mouse
Arg1	AACACGGCAGTGGCTTTAACC	GGTTTTCATGTGGCGCATTC	mouse
iNOS	CAGCTGGGCTGTACAAACCTT	CATTGGAAGTGAAGCGGTTCG	mouse
IL-13Rα1	CATGGAGGGTACAAGTTGTTTCC	GTTTTGACTCTTACTCTGACTGTGTAGACA	mouse
IL-1β	GCCTTGGGCCTCAAAGGAAAGAATC	GGAAGACACAGATTCCATGGTGAAG	mouse
MMP9	CTGGACAGCCAGACACTAAAG	CTCGCGGCAAGTCTTCAGAG	mouse
Luciferase	GCGCGGAGGAGTTGTGTT	TCTGATTTTTCTTGCGTCGAGTT	mouse

## Data Availability

Data are contained within the article or Appendix A and may be accessible upon reasonable request to A.I.

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
