# Peer review of "Possible Metastatic Stage-Dependent ILC2 Activation Induces Differential Functions of MDSCs through IL-13/IL-13Rα1 Signaling during the Progression of Breast Cancer Lung Metastasis"

_cancers, 2022, doi:10.3390/cancers14133267_

Round 1
Reviewer 1 Report
I think the paper is interesting, but the authors should introduce the recent technology to evaluate lung metastasis based on the tumor microenvironment in vitro and in vivo. It is a little difficult to understand the novelty.
Why were the mice models selected? The authors should introduce the other cancer models for invasion or metastasis.
In vitro
Review
Cancers 2020, 12(10), 2754
Research
Acta Biomaterialia 91 (2019) 195
Tissue Eng. Part C Methods 2019, 25, 711–720 https://doi.org/10.1089/ten.tec.2019.0189
In vivo
Review doi.org/10.1093/carcin/bgh261
Research
Nature volume 527, pages472–476 (2015)
Author Response
Responses to Reviewer’s comments
We are grateful for the reviewers’ thoughtful comments and suggestions. As outlined below, we have addressed and/or clarified each of the issues in a point-by-point fashion. We believe these changes have greatly improved the quality and significance of our study and that the manuscript is now suitable for publication. (Please note that the corrections/changes throughout the revised manuscript have been highlighted in yellow, and page and line numbers have been added for your convenience.)
Reviewer 1
Comment 1: I think the paper is interesting, but the authors should introduce the recent technology to evaluate lung metastasis based on the tumor microenvironment in vitro and in vivo. It is a little difficult to understand the novelty.
Response 1: We sincerely appreciate the reviewer’s thoughtful suggestion. Accordingly, we have added a discussion on the recent technological advances, both in vitro and in vivo affecting the research of cancer metastasis (page 15, lines 573-584). These advanced may further our understanding of the interaction between ILC2s and MDSCs in lung metastasis. We believe the novelty of our study lies in the snapshot we have captured of immune cell interactions during the dynamic transition from micro- to macro-metastasis at the 3-week time point following tumor inoculation.
Comment 2: Why were the mice models selected? The authors should introduce the other cancer models for invasion or metastasis.
Response 2: We thank the reviewer for raising this important question. Certainly, we need to study this ILC2-MDSC interaction using another metastatic mouse model. However, it was not possible to perform such an experiment within the limited time frame (i.e., submission of the revised manuscript). The reason for using 4T1/LM4 breast cancer model is that it is reproducible in the formation of primary mammary tumor and lung metastases, thus wasting fewer animals. In addition, since the micrometastatic process activates to form macrometastasis over time, we could assess this entire evolution at a defined time point of 3 weeks after tumor inoculation. In accordance with the reviewer’s suggestion, we have described the other cancer models for invasion or metastasis in the Discussion section (page 15, lines 582-584).

Reviewer 2 Report
The publication "Possible metastatic stage-dependent ILC2 activation induces differential functions of MDSCs through IL-13 / IL-13Rα1 signaling during the progression of breast cancer lung metastasis" is an interesting scientific study. The obtained and developed research results are also interesting from the clinical point of view. The publication is edited in a typical and careful manner. The Materials and Methods and Results sections are elaborated in detail. Figure 4 is completely illegible and should be increased in size. It should be emphasized that the Authors placed figure 6, which is a graphic abstract. References are selected appropriately and are up-to-date, as the most of them have been published in the last 10-15 years. Due to the scientific value and the potential prognostic aspect of the study "Possible metastatic stage-dependent ILC2 activation induces differential functions of MDSCs through IL-13 / IL-13Rα1 signaling during the progression of breast cancer lung metastasis", I propose to publish it in CANCERS in its present form.
Author Response
Responses to Reviewer’s comments
We are grateful for the reviewers’ thoughtful comments and suggestions. As outlined below, we have addressed and/or clarified each of the issues in a point-by-point fashion. We believe these changes have greatly improved the quality and significance of our study and that the manuscript is now suitable for publication. (Please note that the corrections/changes throughout the revised manuscript have been highlighted in yellow, and page and line numbers have been added for your convenience.)
Reviewer 2
Comment 1: The publication "Possible metastatic stage-dependent ILC2 activation induces differential functions of MDSCs through IL-13 / IL-13Rα1 signaling during the progression of breast cancer lung metastasis" is an interesting scientific study. The obtained and developed research results are also interesting from the clinical point of view. The publication is edited in a typical and careful manner. The Materials and Methods and Results sections are elaborated in detail. Figure 4 is completely illegible and should be increased in size. It should be emphasized that the Authors placed figure 6, which is a graphic abstract. References are selected appropriately and are up-to-date, as the most of them have been published in the last 10-15 years. Due to the scientific value and the potential prognostic aspect of the study "Possible metastatic stage-dependent ILC2 activation induces differential functions of MDSCs through IL-13 / IL-13Rα1 signaling during the progression of breast cancer lung metastasis", I propose to publish it in CANCERS in its present form.
Response 1: We thank the reviewer for his/her appreciation of the scientific value our findings. Regarding the reviewer’s suggestion, we have enlarged the image shown in Figure 5 (page 10). We have also changed Figure 3E to Figure 4 (page 9), so that the original Figure 4 is now Figure 5.

Reviewer 3 Report
This paper from Ito et al. deals with immune-microenvironment of breast cancer lung metastasis with a focus on the role of ILC2 in triggering MDSCs' immuno-suppressive function. In particular, the authors set up a protocol, for in vivo experiments, to distinguish between early and late stages of lung metastatization.
In general, the paper is well written, in particular ABSTRACT, INTRODUCTION and MATERIALS AND METHODS are clear and properly explained and exhaustive. Nevertheless, results seem poorly deepened and conclusions are often overstated.
The following comments are meant to improve the paper and make it more robust and comprehensible.
Major comments:
1- lines 223-224: if 20% of mice die before the end-point chosen (21 days), why authors haven’t anticipate the end-point? This protocol results in the loss of “materials” (mice, thus samples) and actually raise questions about the correct application of the procedure
2- DISCUSSION is way too long, it should not be a recap of the results. There is much speculation often not completely supported by presented results. In particular, there are some very serious issues:
- line 467: the presented data do not allow to distinguish whether the release of IL-33 from metastasis in paracrine while the release from the primary tumor is systemic. The only way to say this for sure is because other work has already demonstrated it (references 19 and 48) as cited by authors in lines 495-498. Thus, the presented results are not novel.
- lines 578-579: the protocol presented in this study for separate micro and macro metastasis in mice is the key point of the research. If authors claim that it is not accurate, then the whole research is not reliable
- lines 585-590: here authors describe other works where another protocol for separate micro and macro metastasis were used and state that this technique is more effective than theirs. So, why do authors not use this procedure? This is a very critical and serious point
- lines 593-595: this statement is obviously wrong because of what the authors state in lines 580-582
- lines 604-619:the functional characterization of the ILC2-MDSC axis is based on very weak data. A more in-depth analysis should be performed to connect the authors’ results and literature in an attempt to provide some new information regarding this topic without overstating
Minor comments:
1- line 234: “histological observation” is cited but no such type of results is shown. Is this a typo?
2- lines 249-252: all these details are not suitable for a figure legend
3- line 256: “…effect of metastatic microenvironment on ILC2…”, the point is not the contrary?
4 -line 314: authors have to clearly state that this difference is not statistically significant
5- line 315: Supp Fig S4 shows immunofluorescence, not immunohistochemical staining
6- lines 317-319: this conclusion is unclear and probably overstated, in particular, the connection with PD1 levels which, in addition, did not change in a statistically significant manner
7- line 324: what does “(through IL-5)” mean?
8- figure 3E: please carefully check statistics. Some of the differences defined as significant (*) have great standard deviation
9- lines 336-340: all these details are not suitable for a figure legend
10- lines 353-354: authors have to clearly state that this difference is not statistically significant
11- lines 368-369: the conclusion of this chapter (3.4) is unclear and based on no statistically significant differences In Figure 4A it is not explained the difference between the 3 rows of IF images.
12- lines 400-402: this is an overstatement, the role of PD1 is unclear
13- lines 448-457: all these details are not suitable for a figure legend
Author Response
Responses to Reviewer’s comments
We are grateful for the reviewers’ thoughtful comments and suggestions. As outlined below, we have addressed and/or clarified each of the issues in a point-by-point fashion. We believe these changes have greatly improved the quality and significance of our study and that the manuscript is now suitable for publication. (Please note that the corrections/changes throughout the revised manuscript have been highlighted in yellow, and page and line numbers have been added for your convenience.)
Reviewer 3
Major comments
Comment 1: lines 223-224: if 20% of mice die before the end-point chosen (21 days), why authors haven’t anticipate the end-point? This protocol results in the loss of “materials” (mice, thus samples) and actually raise questions about the correct application of the procedure
Response 1: We acknowledge the reviewer’s point about selection bias, since our analysis does omit severe cases concerning that died prior to the 21-day evaluation date. However, other previous studies using a 4T1 cell line in a breast cancer lung metastasis model (PMID 15240548, 15978719, 15627887) similarly set the evaluation date to 3 weeks after tumor cell inoculation. Therefore, we chose this timing to align comparisons of disease severity to conform to what has been a standard study protocol involving a lung metastasis mice model.
Comment 2: line 467: the presented data do not allow to distinguish whether the release of IL-33 from metastasis in paracrine while the release from the primary tumor is systemic. The only way to say this for sure is because other work has already demonstrated it (references 19 and 48) as cited by authors in lines 495-498. Thus, the presented results are not novel.
Response 2: We acknowledge the reviewer’s comment that the result is not novel since it has already been reported that IL-33 is released from the lung and primary tumor in a mouse model of breast cancer metastasis to the lung. Therefore, it is true to say that there is no novelty with respect to the expression of IL-33 in primary tumors and the metastasized lung. We have modified this sentence in the Discussion section accordingly (page 13, lines 451-453).
Comment 3: - lines 578-579: the protocol presented in this study for separate micro and macro metastasis in mice is the key point of the research. If authors claim that it is not accurate, then the whole research is not reliable
Response 3: We agree with the reviewer’s assertion that if our method of separating the micro- and macro-metastasis regions is not accurate, then the entire research plan would not be reliable.
However, we showed the difference in Luciferase expression between micro- and macro-metastasis in Figure 1D. Since 4T1/LM4 cells contain Luciferase, this difference implies a difference of tumor volume. Furthermore, although we acknowledge the limitation of a few technical ambiguities, we believe the value of our study is that we have captured a snapshot of immune cell interactions in the process from micro- to macro-metastasis at the time point of 3 weeks after tumor inoculation.
Comment 4: - lines 585-590: here authors describe other works where another protocol for separate micro and macro metastasis were used and state that this technique is more effective than theirs. So, why do authors not use this procedure? This is a very critical and serious point
Response 4: We sincerely appreciate the reviewer’s thoughtful comments. Certainly, we need to conduct similar studies of time-course dependent separation for early or late-stage metastasis in the future. These head-to-head experiments are very important. However, the resources we have available for this study were limited. In addition, our study aimed to obtain snapshots of temporal changes in immune cell interactions during the micro- to macro-metastatic stages at a time point of 3 weeks after tumor inoculation. Therefore, we believe that the methodology we used in this study is consistent with our research objectives.
Comment 5: - lines 593-595: this statement is obviously wrong because of what the authors state in lines 580-582
Response 5: We agree with the reviewer’s contention. Therefore, we have excluded this sentence from the Discussion section.
Comment 6: - lines 604-619: the functional characterization of the ILC2-MDSC axis is based on very weak data. A more in-depth analysis should be performed to connect the authors’ results and literature in an attempt to provide some new information regarding this topic without overstating.
Response 6: We acknowledge the reviewer’s comment contending that the functional characterization of the ILC2-MDSC axis is based on very weak data. Hence, we have replaced the phrase “ILC2-MDSC axis” with the phrase “potential interaction of ILC2 with MDSC”. However, previous studies (PMID 34217145) have confirmed that the transfer of ILC2s induces MDSC proliferation and MDSC-derived immunosuppressive cytokines in a 4T1 breast cancer mice model, and have referred to the ILC2-MDSC axis. Furthermore, in human bladder cancer, a positive correlation has also been shown between ILC2 and MDSC, which has been linked to bladder cancer prognosis (PMID 28650339). We need to conduct a further study using the methods of these previous studies. However, we did not have the resources to do so within this short frame (i.e., submission of the revised manuscript). Therefore, we have corrected any overstatements based on the results of our study - see the Introduction section (page 3, lines 109-111), Discussion section (page 15, lines 587-590).
Minor comments
Comment 1: - line 234: “histological observation” is cited but no such type of results is shown. Is this a typo?
Response 1: We thank the reviewer for drawing our attention to this error. This was a typo and has been corrected (page 6, line 255).
Comment 2: - lines 249-252: all these details are not suitable for a figure legend
Response 2: We agree with reviewer’s comment and have incorporated this suggestion throughout our manuscript. In accordance with the reviewer’s comment, we have added a sentence regarding this statistical method in the Materials and Methods section (page 5, lines 216-219).
Comment 3: - line 256: “…effect of metastatic microenvironment on ILC2…”, the point is not the contrary?
Response 3: We agree with reviewer’s comment and have correct this error. In accordance with the reviewer’s comment, we have corrected the various grammatical issues noted in the manuscript.
Comment 4: -line 314: authors have to clearly state that this difference is not statistically significant
Response 4: We agree with reviewer’s comment and have correct this. Specifically, we have added the statement that “, but not statistically significant.” (page 8, lines 313-314).
Comment 5: -line 315: Supp Fig S4 shows immunofluorescence, not immunohistochemical staining
Response 5: We agree with reviewer’s comment and have correct this error (page 8, line 314).
Comment 6: - lines 317-319: this conclusion is unclear and probably overstated, in particular, the connection with PD1 levels which, in addition, did not change in a statistically significant manner
Response 6: We agree with reviewer’s comment that PD-1 expression on ILC2 is unclear and is probably overstated.Therefore, we have modified this sentence accordingly (page 8, lines 317-320).
Comment 7: - line 324: what does “(through IL-5)” mean?
Response 7: This was spelled wrong and has been corrected (page 9, line 336).
Comment 8: - figure 3E: please carefully check statistics. Some of the differences defined as significant (*) have great standard deviation.
Response 8: We thank the reviewer for raising this point. Accordingly, we have changed Figure 3E to Figure 4 (page 9). Since some of the figures (Figure 4, Figure 6c, Supplemental Fig S6, Supplemental Fig S7) were presented as mean ± SD, we have corrected them to mean ± SEM. These corrections did not affect the statistical significance.
Comment 9: -lines 336-340: all these details are not suitable for a figure legend.
Response 9: We agree with reviewer’s comment and have incorporated this suggestion throughout our manuscript – see the Materials and Methods section (page 5, lines 216-219).
Comment 10: - lines 353-354: authors have to clearly state that this difference is not statistically significant
Response 10: We agree with reviewer’s comment and have corrected it to “, but not statistically significant.” (page 8, lines 313-314).
Comment 11: - lines 368-369: the conclusion of this chapter (3.4) is unclear and based on no statistically significant differences In Figure 4A it is not explained the difference between the 3 rows of IF images.
Response 11: We agree with the reviewer’s comment that chapter 3.4 is unclear and the finding not statistically significant. We have changed Figure 3E to Figure 4, so that the original Figure 4 is now Figure 5. In accordance with the reviewer’s comment, we have modified the sentence noted in the manuscript (page 10, lines 364-368).
Comment 12: - lines 400-402: this is an overstatement, the role of PD1 is unclear.
Response 12: We sincerely appreciate the reviewer’s thoughtful comments. Accordingly, we have modified the relevant sentence in the manuscript (page 11, lines 400-402).
Comment 13: - lines 448-457: all these details are not suitable for a figure legend.
Response 13: We agree with the reviewer’s comment and have excluded these sentences accordingly.

Reviewer 4 Report
The paper submitted to cancers MDPI, by Atsushi Ito and co-workers, titled: “Possible metastatic stage-dependent ILC2 activation induces differential functions of MDSCs through IL-13/IL-13Rα1 signaling during the progression of breast cancer lung metastasis” presents an extensive study of breast cancer metastasis to lung in context of immune cells infiltration, expansion and activation.
The authors focus on highly metastatic syngeneic model of breast cancer using derivative from 4T1 luciferin expressing cell line that seeds breast cancer in lung upon syngeneic primary tumor implantation.
Authors focus on two populations of immune cells: ILC2 and MDSC monocytic subpopulation that previously was described to support metastasis development and progression. The main weakness of publication is a model that is based on a single cell line at a single time point (21 days post-injection) assessing different stages of metastases progression in lung via macroscopic evaluation and tissue excision.
Authors differentiate micro from macro metastases in lung, and compare isolated immune cells with healthy lung tissue. Immune cells analysis includes several specific markers for ILC2s, MDSCs, intracellular cytokines and immune cells activation markers. The paper presents results on IL-13 and several other differences in micro versus macro metastases with conclusions that tumor stroma supports growth of metastasis in lungs. The paper requires more careful data analysis, including additional controls and text clarifications. Therefore major revisions are recommended prior publication, including additional experimental controls and text re-organization with editorial corrections:
- In Methods section under, “2.2. Development of mouse 4T1/LM4 breast cancer model”, authors describe breast cancer implantation and development of mets including luciferin positive breast cancer cell line. At 21 days post-implant tumors and lungs were harvested and lungs were weighted, photographed and assessed for mets: 2mm and larger tumor nodules were excised from lungs and considered macro-mets. However there is no clarification how micro-mets were evaluated. Was it the remaining lung tissue? Did micro mets were excised from different lobes than macro-mets?
- Authors prepared lung tissue sections for immunofluorescence presented in Figure 4 with IL-33 staining. It is recommended to present either as a luciferase expression ex vivo or as a microscopical IHC picture and show the size of micro- versus macro mets in the lung. The 4T1/LM4 cells already contain luciferase, so the ex vivo pictures of isolated micro- and macro mets would help to properly categorize mets.
- It would be helpful to have a microscopic pic with stained breast cancer specific marker in lung mets showing differences between micro and macro-mets.
- In the Methods section “Isolation of leucocytes from lungs”- authors normalize the number of leucocytes in micro- macro-mets to the wet weight of the samples. Is there other way of normalizing the number of cells to the total number of leucocytes isolated from the lung, or to the number of tumor cells, expressed as a luciferase???
- In the Methods section under “RNA isolation and real-time quantitative PCR”, authors briefly describe the isolation of total RNA from tissues and use of RT-qPCR to assess gene expression. Please elaborate here on specific gene expression and gene normalization method used in the specific Figures. What was the housekeeping gene expression for tumor cells, versus MDSC or ILC2, or cytokines genes. Please list and differentiate the housekeeping genes depending on the population of cells used for analysis.
- In Figure 1B authors show bar graph representing tumor growth, this corresponds to the two time points measured at 14 days and 21 days post-injection, however in Methods section authors say that the tumors were measures every 3 days post-injection. Please provide the full tumor growth curves or modify the description in Methods section.
- Supplementary Figure 4 is lacking picture of DAPI, even though it is describing DAPI staining, as additional channel for the overlay.
- In Figure 3E authors need to carefully introduce tissue that is being investigated for IL-13, IL-5, IL-1b and MMP-9 gene expression and please specify the loading control, or housekeeping gene, used for normalization and quantitation by RT-qPCR. In subsequent paragraph: “3.3. ECM remodeling programs are induced during micro-metastasis” authors describe Figure 3E, and point to differences between genes expression in micro-metastases, versus macro metastases. Figure 3A, B, C and D panels present data from isolated immune cells, ILCs, from mice lungs. However, Figure 3E is showing data from normal lung, micro and macro- mets, which is slightly inconsistent with Figure 3 title and its purpose. Figure 3E requires re-arrangement.
- In Results section 3.4 authors state: “Fluorescence intensity of IL-33 was elevated not only in metastatic sites, but also in the primary tumors.” However, in Supplementary Figure 6 authors include data from RT-qPCR that does not support increased IL-33 gene expression level in metastatic lungs isolated from 4T1/LM4 mice. Authors conclude in the Discussion, verse 493-495, that IL-33 is produced by primary tumor and travels through bloodstream reaching lungs. It may not be a reasonable explanation, since Supplementary Fig 6, for gene expression authors used as a housekeeping gene, Actb, which is not specific enough for the 41T/LM4 mets, but specific to the total mRNA isolated and includes normal mouse lung tissue. The loading control should be specific to 4T1/LM4, such as luciferase gene, as this will provide more adequate measure for IL-33 production in mice lungs by metastatic lesions. In addition, it is recommended to use immune system specific control for gene expression normalization where applicable (eg. Figure 5C).
- Throughout the text in the legends of Figures: 1, 3, 4, and 5, authors provide the description of statistical analysis as ordinary one-way Anova followed by multiple comparisons test by a “Turkey”à a test is misspelled as it is “Tukey”
- Figure 5C with quantitative gene expression assessed in sub-populations of PMN-MDSC and M-MDSC requires detailed description, including a specific housekeeping genes used for data normalization. Was the same housekeeping gene used for both PMN-and M-MDSC?
- Figure 4 needs to be revised and fluorescent microscopy pictures require correct names and titles to assess differences between the provided tissues.
Author Response
Responses to Reviewer’s comments
We are grateful for the reviewers’ thoughtful comments and suggestions. As outlined below, we have addressed and/or clarified each of the issues in a point-by-point fashion. We believe these changes have greatly improved the quality and significance of our study and that the manuscript is now suitable for publication. (Please note that the corrections/changes throughout the revised manuscript have been highlighted in yellow, and page and line numbers have been added for your convenience.)
Reviewer 4
Comment 1: In Methods section under, “2.2. Development of mouse 4T1/LM4 breast cancer model”, authors describe breast cancer implantation and development of mets including luciferin positive breast cancer cell line. At 21 days post-implant tumors and lungs were harvested and lungs were weighted, photographed and assessed for mets: 2mm and larger tumor nodules were excised from lungs and considered macro-mets. However there is no clarification how micro-mets were evaluated. Was it the remaining lung tissue? Did micro mets were excised from different lobes than macro-mets?
Response 1: We thank the reviewer for raising this important question. Micrometastatic region sample contain the lung tissues of normal regions after eliminating macroscopically detectible tumors in the same lobes. Thus, this was the remaining lung tissues after tumor resection and we acknowledge the presence of normal tissues in the micrometastatic samples could induce any bias, which we have discussed in the Discussion section (Page 15, lines 551-570). For a more detailed explanation, we have added a sentence in the Materials and Methods section (page 3, lines 144-146).
Comment 2: Authors prepared lung tissue sections for immunofluorescence presented in Figure 4 with IL-33 staining. It is recommended to present either as a luciferase expression ex vivo or as a microscopical IHC picture and show the size of micro- versus macro mets in the lung. The 4T1/LM4 cells already contain luciferase, so the ex vivo pictures of isolated micro- and macro mets would help to properly categorize mets.
Response 2: We agree with the reviewer’s thoughtful suggestion. As he/she mentioned, it is better to show the ex vivopictures of isolated micro- and macro-metastasis regions by luminometer using the Luciferase assay system. However, it was not possible to perform the experiment within the limited time frame (i.e., submission of the revised manuscript). Moreover, we do not have the resources to show them now. Therefore, we will address your important suggestion in a future study.
Comment 3: It would be helpful to have a microscopic pic with stained breast cancer specific marker in lung mets showing differences between micro and macro-mets.
Response 3: We agree with the reviewer’s thoughtful suggestion. However, it was not possible to perform the experiment within the limited time frame (i.e., submission of the revised manuscript). Moreover, we do not have the resources to show them now. Therefore, we will address your important suggestion in a future study.
Comment 4: In the Methods section “Isolation of leucocytes from lungs”- authors normalize the number of leucocytes in micro- macro-mets to the wet weight of the samples. Is there other way of normalizing the number of cells to the total number of leucocytes isolated from the lung, or to the number of tumor cells, expressed as a luciferase???
Response 4: We thank the reviewer for raising this important question. We counted the total number of leucocytes from the lung after using the Percoll gradient isolation method. The percentage of ILC2 in total leucocytes was then measured by flow cytometry. Our method of dividing the metastasized lung into micro- and macro-metastasis required the normalization for comparisons due to the different tissue weights. Unfortunately, we could not establish any other way than normalizing by tissue weight. However, another study evaluating the role of ILC2 in pancreatic cancer (PMID 34099918) also normalized by tumor tissue weight. Therefore, we believe our method to be appropriate.
Comment 5: In the Methods section under “RNA isolation and real-tme quantitative PCR”, authors briefly describe the isolation of total RNA from tissues and use of RT-qPCR to assess gene expression. Please elaborate here on specific gene expression and gene normalization method used in the specific Figures. What was the housekeeping gene expression for tumor cells, versus MDSC or ILC2, or cytokines genes. Please list and differentiate the housekeeping genes depending on the population of cells used for analysis.
Response 5: We sincerely appreciate the reviewer’s thoughtful suggestion. We have listed the Primer sequences used for the real-time qPCR analysis in the Materials and Methods section (page 5, lines 201-202). We used β-actin ford housekeeping gene expression and normalized mRNA levels using the 2-ΔΔCT method for all real-time qPCR analyses.
Comment 6: In Figure 1B authors show bar graph representing tumor growth, this corresponds to the two time points measured at 14 days and 21 days post-injection, however in Methods section authors say that the tumors were measures every 3 days post-injection. Please provide the full tumor growth curves or modify the description in Methods section.
Response 6: We thank the reviewer for drawing our attention to these issues. We stated that the tumors were measured once every 2 to 3 days. Therefore, we have modified the description in the Methods section (page 3, line 132) and in the Results section (page5, line 226).
Comment 7: Supplementary Figure 4 is lacking picture of DAPI, even though it is describing DAPI staining, as additional channel for the overlay.
Response 7: We agree with reviewer’s comment and have incorporated this suggestion throughout our manuscript. Specifically, we have added the DAPI staining picture of a tumor-free lung and 4T1/LM4 lung to Supplemental Figure S4.
Comment 8: In Figure 3E authors need to carefully introduce tissue that is being investigated for IL-13, IL-5, IL-1b and MMP-9 gene expression and please specify the loading control, or housekeeping gene, used for normalization and quantitation by RT-qPCR. In subsequent paragraph: “3.3. ECM remodeling programs are induced during micro-metastasis” authors describe Figure 3E, and point to differences between genes expression in micro-metastases, versus macro metastases. Figure 3A, B, C and D panels present data from isolated immune cells, ILCs, from mice lungs. However, Figure 3E is showing data from normal lung, micro and macro- mets, which is slightly inconsistent with Figure 3 title and its purpose. Figure 3E requires re-arrangement.
Response 8: We sincerely appreciate the reviewer’s thoughtful comments. We have renumbered Figure 3E as Figure 4 accordingly and have revised the figure title (page 9, line 355).
Comment 9: In Results section 3.4 authors state: “Fluorescence intensity of IL-33 was elevated not only in metastatic sites, but also in the primary tumors.” However, in Supplementary Figure 6 authors include data from RT-qPCR that does not support increased IL-33 gene expression level in metastatic lungs isolated from 4T1/LM4 mice. Authors conclude in the Discussion, verse 493-495, that IL-33 is produced by primary tumor and travels through bloodstream reaching lungs. It may not be a reasonable explanation, since Supplementary Fig 6, for gene expression authors used as a housekeeping gene, Actb, which is not specific enough for the 41T/LM4 mets, but specific to the total mRNA isolated and includes normal mouse lung tissue. The loading control should be specific to 4T1/LM4, such as luciferase gene, as this will provide more adequate measure for IL-33 production in mice lungs by metastatic lesions. In addition, it is recommended to use immune system specific control for gene expression normalization where applicable (eg. Figure 5C).
Response 9:
We sincerely appreciate the reviewer’s thoughtful comments. As a possible reason for the discrepancy between the fluorescence intensity and gene expression of IL-33 in metastases-bearing mice, we speculated that the IL-33 produced by the primary mammary tumor may reach the lungs through the bloodstream. Therefore, we compared the concentrations of serum IL-33 between tumor-free mice and tumor-bearing mice via ELISA using serum samples. Unfortunately, we could not show a significant difference, but serum IL-33 concentrations in tumor-bearing mice tended to be higher than those in tumor-free mice. We have added these results to the Discussion section (page 13, lines 487-489) and to Supplemental Fig. S6C.
In addition, we performed the additional RT-qPCR analysis of IL-33 between three groups: normal lung, metastases-bearing lung and primary mammary tumor using other housekeeping genes (18S ribosomal RNA, β2microgloblin and Hypoxanthine phosphoribosyltransferase: HPRT). However, the results were similar to the analysis using β-actin, with higher IL-33 expression in primary mammary tumors and no significant difference between normal lungs and metastases-bearing lungs.
Based on the reviewer’s suggestion, we compared IL-33 expression between metastases-bearing lungs and primary mammary tumors using the luciferase gene as a loading control. The results showed that IL-33 expression was higher in the metastases-bearing lungs than in the primary tumors (Figure S6B). However, this result requires some interpretation: a previous study using a 4T1 mouse model (PMID 26942079) showed that 4T1 cells are not the main source of IL-33 within tumor tissues. According to this study, IL-33 is produced from epithelial cells, endothelial cells and tumor-associated fibroblasts within the tumor microenvironment. In addition, luciferase expression was lower than in the metastases-bearing lungs than the primary mammary tumors. Therefore, it may not be appropriate to use 4T1/LM4 cell-specific genes as a loading control. We have added these results and consideration to the Discussion section (page 13, lines 478-487) and to Supplemental Fig. S6B.
Comment 10: Throughout the text in the legends of Figures: 1, 3, 4, and 5, authors provide the description of statistical analysis as ordinary one-way Anova followed by multiple comparisons test by a “Turkey”à a test is misspelled as it is “Tukey”
Response 10: We did spell this wrong have corrected it throughout the manuscript accordingly (page5, line 218).
Comment 11: Figure 5C with quantitative gene expression assessed in sub-populations of PMN-MDSC and M-MDSC requires detailed description, including a specific housekeeping genes used for data normalization. Was the same housekeeping gene used for both PMN-and M-MDSC?
Response 11: We thank the reviewer for raising this important question. We used β-actin as a housekeeping gene for normalization in all of RT-qPCR analyses of both PMN- and M-MDSC.
Comment 12: Figure 4 needs to be revised and fluorescent microscopy pictures require correct names and titles to assess differences between the provided tissues.
Response 12: We sincerely appreciate the reviewer’s thoughtful suggestions. Accordingly, we have enlarged the image in Figure 5 and have revised the title to describe the differences between samples. We have also changed Figure 3E to Figure 4; thus, the original Figure 4 is now Figure 5.

Round 2
Reviewer 3 Report
I would like to thank the authors for their willingness and their efforts to improve this paper. Anyway, some important issues remain.
Response 1: cited papers (PMID 15240548, 15978719, 15627887) should be implemented in the final version of the work.
Response 4: I profoundly understand the difficulties authors claim about ("the resources we have available for this study were limited"). Unfortunately, in this context, the study has to be evaluated for what it presents and it has to stand alone. For this purpose, the evident lack of novelty in the methodology is critical. The little differences in previously reported methodology are not enough.
Response 5: I am not sure that these sentences have really been deleted in the new version of the paper. By the way, simply deleting these sentences is not sufficient. The point is that authors honestly acknowledge the limitations of their methodology but they try anyway to draw conclusions which sound doubtful.
Authors claim to have deleted the sentence I was concerned about: "We believe that our results have captured a snapshot of the dynamic process that occurs during the progression from micro- to macrometastasis" (lines 593-595, old manuscript version).
First, this is not true because this sentence is still present in the revised manuscript (lines 570-572, new manuscript version).
Second, the point is not removing a sentence, but the concept behind it. The presented methodology lack of novelty and clear efficacy and authors acknowledge these weaknesses (in the sentence "We compared micro- and macrometastasis at the same time during the late stage, which have resulted in tumorigenesis related bias: regions that remained micrometastatic until the late stage may simply reflect slow tumor growth due to unfavorable TME.", lines 579-581 old version, lines 556-558 new version). Anyway, they speculate on these results and discuss them as they have no flaws. In my opinion, this is not acceptable for a paper to be published on Cancers. There is no way to fix this point.
Author Response
Responses to Reviewers’ comments
We are grateful for the reviewers’ thoughtful comments and suggestions. As outlined below, we have addressed and/or clarified each of the issues in a point-by-point fashion. We believe these changes have greatly improved the quality and significance of our study and that the manuscript is now suitable for publication. (Please note that the corrections/changes throughout the revised manuscript have been highlighted in yellow, and page and line numbers have been added for your convenience.)
Reviewer 3 
Comment 1: Response 1: cited papers (PMID 15240548, 15978719, 15627887) should be implemented in the final version of the work.
Response 1: We agree with reviewer’s comment and have incorporated this suggestion throughout our manuscript – see the Material and Methods section (page3, lines 134-135).
Comment 2: Response 4: I profoundly understand the difficulties authors claim about ("the resources we have available for this study were limited"). Unfortunately, in this context, the study has to be evaluated for what it presents and it has to stand alone. For this purpose, the evident lack of novelty in the methodology is critical. The little differences in previously reported methodology are not enough.
Response 2: We agree with the reviewer’s assertion about the limitations of our method in separating micro- and macrometastasis. In this context, we do not intend to claim the novelty and superiority of our method. Rather, we believe that our method shares an important limitation with that of previous studies in that both methods only capture a snapshot of a certain time point in the progression of breast cancer lung metastasis. However, there are some differences in the interpretation of the study results. For example, our methods may be susceptible to a tumorigenesis-related bias: regions that remained micrometastatic until the late stage may simply reflect slow tumor growth due to unfavorable TME. In addition, it is not certain when the micrometastases were formed in the lung during these late stages. On the other hand, the method employed by previous studies may not exclude the influence of the micrometastatic regions present in the whole lung during the late stage, which is defined as macrometastasis. However, the snapshot approach used by the present and previous studies cannot unravel the dynamics underlying the metastatic cascade. In accordance with the reviewer’s comments, we have modified the relevant sentences in the manuscript (pages 15-16, lines 566-587).
Comment 3: Response 5: I am not sure that these sentences have really been deleted in the new version of the paper. By the way, simply deleting these sentences is not sufficient. The point is that authors honestly acknowledge the limitations of their methodology but they try anyway to draw conclusions which sound doubtful.
Authors claim to have deleted the sentence I was concerned about: "We believe that our results have captured a snapshot of the dynamic process that occurs during the progression from micro- to macrometastasis" (lines 593-595, old manuscript version).
First, this is not true because this sentence is still present in the revised manuscript (lines 570-572, new manuscript version).
Second, the point is not removing a sentence, but the concept behind it. The presented methodology lack of novelty and clear efficacy and authors acknowledge these weaknesses (in the sentence "We compared micro- and macrometastasis at the same time during the late stage, which have resulted in tumorigenesis related bias: regions that remained micrometastatic until the late stage may simply reflect slow tumor growth due to unfavorable TME.", lines 579-581 old version, lines 556-558 new version). Anyway, they speculate on these results and discuss them as they have no flaws. In my opinion, this is not acceptable for a paper to be published on Cancers. There is no way to fix this point.
Response 3: We apologize for not having made the needed corrections to the points raised by the reviewer, although this error stems from the fact that the line numbers of the manuscript were changed. We are grateful to the reviewer for answering this question about the corresponding line numbers in the manuscript. We also acknowledged the reviewer’s concerns about the lack of novelty and clear efficacy of the presented methodology. As we mentioned in Comment 2, our intention is not to emphasize the novelty and superiority of our method. We would like to mention that there is some room for different interpretations and implications of the results obtained by our method, and have discussed them in the Discussion section. In accordance with the reviewer’s thoughtful comments, we have modified the limitations part of the Discussion section – see pages 15-16, lines 566-587.

Reviewer 4 Report
In the Materials and Methods authors need to introduce ELISA as additional method used for evaluation of IL-33 concentration in mice serum.
In addition Figure 5 presenting immunofluorescence pics is missing correct description side by side with pictures showing respectively: DAPI, Cy3-IL33 vs overlay. It needs to be introduced in Figure, what picture represents which staining.
Author Response
Responses to Reviewers’ comments
We are grateful for the reviewers’ thoughtful comments and suggestions. As outlined below, we have addressed and/or clarified each of the issues in a point-by-point fashion. We believe these changes have greatly improved the quality and significance of our study and that the manuscript is now suitable for publication. (Please note that the corrections/changes throughout the revised manuscript have been highlighted in yellow, and page and line numbers have been added for your convenience.)
Reviewer 4
Comments and Suggestions for Authors
Comment 1: In the Materials and Methods authors need to introduce ELISA as additional method used for evaluation of IL-33 concentration in mice serum.
Response 1: We agree with reviewer’s comment and have incorporated this suggestion throughout our manuscript – see the Material and Methods section (page5, lines 215-222).
Comment 2: In addition Figure 5 presenting immunofluorescence pics is missing correct description side by side with pictures showing respectively: DAPI, Cy3-IL33 vs overlay. It needs to be introduced in Figure, what picture represents which staining.
Response 2: We sincerely appreciate the reviewer’s thoughtful comments. Accordingly, we have modified Figure 5A to show DAPI, Cy-IL-33 and Merge.
